# A Role for the WNT Co-Receptor LRP6 in Pathogenesis and Therapy of Epithelial Cancers

**DOI:** 10.3390/cancers11081162

**Published:** 2019-08-13

**Authors:** Jennifer Raisch, Anthony Côté-Biron, Nathalie Rivard

**Affiliations:** Department of Immunology and Cell Biology, Université de Sherbrooke, Sherbrooke, QC J1E 4K8, Canada

**Keywords:** low-density lipoprotein receptor-related protein 6, LRP6, carcinoma, WNT signaling, β-catenin

## Abstract

The WNT/β-catenin signaling pathway controls stem and progenitor cell proliferation, survival and differentiation in epithelial tissues. Aberrant stimulation of this pathway is therefore frequently observed in cancers from epithelial origin. For instance, colorectal and hepatic cancers display activating mutations in the *CTNNB1* gene encoding β-catenin, or inactivating *APC* and *AXIN* gene mutations. However, these mutations are uncommon in breast and pancreatic cancers despite nuclear β-catenin localization, indicative of pathway activation. Notably, the low-density lipoprotein receptor-related protein 6 (LRP6), an indispensable co-receptor for WNT, is frequently overexpressed in colorectal, liver, breast and pancreatic adenocarcinomas in association with increased WNT/β -catenin signaling. Moreover, LRP6 is hyperphosphorylated in KRAS-mutated cells and in patient-derived colorectal tumours. Polymorphisms in the *LRP6* gene are also associated with different susceptibility to developing specific types of lung, bladder and colorectal cancers. Additionally, recent observations suggest that LRP6 dysfunction may be involved in carcinogenesis. Indeed, reducing LRP6 expression and/or activity inhibits cancer cell proliferation and delays tumour growth in vivo. This review summarizes current knowledge regarding the biological function and regulation of LRP6 in the development of epithelial cancers—especially colorectal, liver, breast and pancreatic cancers.

## 1. Introduction

Cell proliferation and differentiation are finely regulated during epithelial tissue development and renewal. One of the most important signaling pathways involved in this regulation is the ubiquitous WNT/β-catenin pathway due to its crucial role in adult stem cell regulation [1]. Accordingly, abnormal WNT/β-catenin signaling pathway activation is therefore involved in the pathogenesis of various human diseases, particularly cancer.

β-catenin protein, encoded by *CTNNB1*, acts as the main signal transducer. In the absence of WNT factors, β-catenin cytoplasmic concentrations are kept low because of active degradation through the proteasome (Figure 1A). Indeed, in the cytoplasm, β-catenin is associated with a “destruction complex” that includes casein kinase 1 (CK1α), glycogen synthase kinase 3 (GSK3), AXIN and adenomatous polyposis coli (APC) proteins. Within this complex, β-catenin is phosphorylated by CK1α at serine 45. This event leads to subsequent phosphorylations on residues 41, 37 and 33 by GSK3, which mark β-catenin for ubiquitin-mediated degradation in the proteasome [2,3]. WNT ligands bind to the seven-pass transmembrane receptor Frizzled (FZD) and the single-pass low-density lipoprotein receptor-related protein 5 or 6 (LRP5/6) (Figure 1B). The WNT–FZD–LRP5/6 trimeric complex recruits the scaffold protein Dishevelled (DVL) which itself polymerizes to bind AXIN [4]. This structure promotes LRP6 aggregation, leading to the formation of LRP6 signalosomes [5]. LRP5/6 is then phosphorylated by both CK1γ and GSK3, releasing free β-catenin in the cytoplasm. LRP6 phosphorylations allow the recruitment of more AXIN and GSK3, acting as an inhibitor of this kinase, amplifying WNT signal activation and allowing sustained activation of the pathway [6]. β-catenin then accumulates in the cytoplasm and translocates in the nucleus, where β-catenin binds to T-cell factor (TCF) or lymphoid enhancer-binding protein factor (LEF). The fully active transcription factor induces the transcription of genes involved in the control of cell cycle, survival and differentiation.

Mutations in WNT/β-catenin pathway coding genes such as *CTNNB1*, *AXIN* and *APC* are found in colorectal and hepatic cancers, as described below, and are known to drive tumorigenesis. However, overexpression of these genes is not reported in all carcinomas, suggesting the involvement of other key components of this pathway in carcinogenesis. In this signaling cascade, coreceptors LRP5/6 are instrumental for downstream signal transduction. Indeed, the expression of non-phosphorylatable LRP5/6 mutants abrogates β-catenin-TCF/LEF transcriptional activity [7]. Herein, we focus on the specific roles of LRP6 in the control of the WNT/β-catenin pathway and in the regulation of tissue homeostasis and carcinoma development.

## 2. LRP6 Structure, Regulation and Role in Tissue Homeostasis

### 2.1. Structure

LRP5 and LRP6 proteins are 70% identical [4]. These receptors consist of an extracellular domain with four YWTD (Tyr–Trp–Thr–Asp)-type β-propeller motifs (P1–P4), four EGF-like domains (E1–E4) and three LDLR type A domains (L1–L3) close to the transmembrane (Figure 2A) [8]. One EGF-like domain following each YWTD motif form together the functional module for ligand recognition which are part of two functional units, namely, P1E1P2E2 and P3E3P4E4. While the P1E1P2E2 unit binds specifically WNT1, WNT2, WNT2b, WNT6 and WNT9b, P3E3P4E4 binds WNT3 and WNT3a [4,8,9,10]. Interestingly, LRP6 dimerization mediated by LDLR domains is required for WNT3a activation of β-catenin signaling [11]. Additionally, LRP6 mutant lacking the extracellular domain (named LRP6ΔN mutant) is a constitutively active form of LRP6 that increases β-catenin-TCF/LEF transcriptional activity. This suggests that the LRP6 extracellular domain exerts a negative control on β-catenin-dependent transcriptional activity, probably due to folding rearrangement (Figure 2B) [12]. In contrast, deletion of the cytoplasmic domain of LRP6 (named LRP6ΔC mutant) induces a dominant-negative effect, abrogating β-catenin-dependent transcriptional activity [13].

### 2.2. Post-Translational Maturation and Regulation

LRP6 maturation involves several post-translational modifications to ensure correct folding. Palmitoylation of Cys-1394 and Cys-1399 residues is essential for LRP6 exit from the endoplasmic reticulum (ER). Indeed, palmitoylation-deficient LRP6 mutants are retained in the ER due to Lys-1403 mono-ubiquitination, resulting in defective WNT/β-catenin pathway activation [15]. Conversely, Lys-1403 mono-ubiquitination is needed for correct folding, allowing LRP6 to interact with a chaperone-acting ubiquitin-binding protein, therefore providing time to correctly fold [16]. If misfolding occurs, LRP6 is poly-ubiquitinated on other lysines and then targeted for ER-associated degradation (ERAD) [16].

Like many membrane-anchored proteins, LRP6 folds inefficiently. LRP6’s folding and membrane targeting require the assistance of chaperones and enzymes. For example, the mesoderm development LRP chaperone (MESD) is a specialized chaperone preventing intermolecular disulfide bond formation in the extracellular domain which induces LRP6 aggregation in the ER [17]. Moreover, the membrane targeting of LRP6 is dependent on the cell-surface glycoprotein CD44 binding to F-actin via Ezrin, suggesting the involvement of actin polymerization in the transport of vesicles [18]. Consequently, CD44 silencing induces abnormal LRP6 folding and ER retention [19].

On the cell surface, the presence of large LRP5/6-containing complexes suggests that the monomeric mode of an FZD-LRP6 coreceptor complex is not privileged. In fact, FZD receptors (FZDs) and LRP5/6 undergo multiple polymerizations on the cell surface upon WNT ligand binding, to form oligomers. WNT ligands selectively bind different states of LRP5/6-FZD receptor oligomerization [20]. For instance, WNT1 subfamily ligands (WNT1, WNT2, WNT2b, WNT6 and WNT9b) preferentially bind receptor complexes with predominant LRP5/6 abundance, while WNT3 subfamily ligands (WNT3 and WNT3a) bind to predominant FZD-containing complexes. Dickkopf-related protein 1 (DKK1) is a secreted protein that antagonizes canonical WNT signaling by competitive binding to the P3E3P4E4 domain of the LRP6 coreceptor. This leads to LRP6 extracellular domain compaction, resulting in the inhibition of both receptor oligomerization and signaling activation [10,14,21]. Conversely, loss of APC—a negative regulator of WNT signaling—induces the formation of large LRP5/6 signalosomes in a ligand-independent manner [20].

Several groups have reported a crucial role for the five proline-rich PPP(S/T)P motifs, containing serine or threonine residues (Ser1490, Thr1530, Thr1572, Ser1590, Ser1607) in LRP6, in the activation of downstream signaling (Figure 2A) [6]. Phosphorylation of these residues by CK1 and GSK3 kinases leads to β-catenin-dependent pathway activation by releasing β-catenin from the destruction complex, and stabilizing β-catenin. When a dominant-negative mutant of CK1γ is expressed, LRP6 phosphorylation is blocked but signalosomes still form, suggesting that LRP6 phosphorylation is a consequence and not the cause of LRP6 aggregation [5]. Mutations of serines or threonines into alanine in the proline-rich motifs also inhibit subsequent binding of AXIN and GSK3 observed during the amplification phase. Therefore, these mutations inhibit sustained activation of the pathway after WNT binding [22,23,24]. In addition to CK1γ and GSK3, LRP6 can be phosphorylated in the proline-rich PPP(S/T)P motifs by mitogen-activated protein kinases ERK, p38 and JNK, resulting in WNT/β-catenin pathway activation [25,26,27]. Finally, direct interaction between the tight junction transmembrane protein BVES/POPDC1 and the LRP6 transmembrane domain leads to LRP6 dephosphorylation by the PP2A phosphatase, thus inducing a negative feedback regulation of WNT signaling pathway stimulation [28]. Therefore, LRP6 may also represent a point of convergence between the WNT pathway and other signaling pathways.

### 2.3. Endocytosis

Endocytosis is required for sustained WNT/β-catenin signal activation. Indeed, endocytosis of the LRP5/6-FZD receptor complex occurs quickly after WNT binding [29]. Interestingly, single amino acid substitutions of LDLR repeat residues inhibit LRP6 internalization and WNT signaling activity. It has been suggested that clathrin- and caveolin-dependent pathways mediate receptor complex endocytosis, depending on the cellular context and the WNT subtype [30,31]. By down-regulating transmembrane receptors, clathrin-mediated endocytosis is frequently involved in signal termination [32]. However, it was recently reported that APC inactivation can induce ligand-independent LRP6 signalosome formation via clathrin-mediated endocytosis [33]. This suggests that, without WNT ligands, APC inhibits receptor activation via the clathrin pathway.

Caveolin-dependent internalization occurs mostly at lipid rafts, a platform for receptor-mediated signaling, which facilitates the sequestration of receptors and signaling molecules within caveolae [34]. In fact, the cytoplasmic tail of LRP6 exhibits two tyrosine-based motifs (YXXY and YXXF) that can constitute an endocytic signal [35]. Mutation of these tyrosines impairs LRP6 endocytosis and induces LRP6 association with caveolin in lipid rafts. It has been proposed that in the absence of WNT signal, LRP6 is distributed between lipid raft and non-lipid-raft domains. Upon binding of WNT glycoproteins, LRP6 and FZD coreceptors are translocated to lipid rafts and aggregate in LRP6 signalosomes [5]. LRP6 is then phosphorylated and internalized by caveolin-dependent mechanisms, an event required for downstream β-catenin pathway activation. Therefore, disruption of the tyrosine-based motifs promotes LRP6 caveolin-mediated endocytosis at the expanse of clathrin-mediated endocytosis [35]. Notably, by phosphorylating these tyrosine residues, SRC and FER kinases reduce LRP6 cell surface distribution and inhibit the WNT/β-catenin pathway after WNT3a stimulation. Thus, LRP6 tyrosine phosphorylation acts as a negative feedback to avoid uncontrolled activation of the WNT pathway [36].

### 2.4. Tissue Homeostasis

#### 2.4.1. Embryonic Development

Mouse models lacking *Lrp6* have been developed in order to determine the role of LRP6 during development. Global *Lrp6* deletion in mice induces a severe phenotype with developmental defects in eye, limb and neural tube [37,38]. Moreover, mice die at birth, indicating a crucial role for LRP6 in embryogenesis, organ formation and function. Notably, the phenotype of *Lrp6* knockout mice is less severe than the phenotype observed in *Ctnnb1* knockout mice, which die during gastrulation, and the phenotype observed in *Wnt3a*, *Wnt1* and *Wnt7a* knockout mice, which also exhibit early embryonic lethality [37,39].

#### 2.4.2. Intestinal Epithelial Development and Homeostasis

The WNT/β-catenin pathway is critical for intestinal epithelial homeostasis and development. Mice overexpressing DKK1, a WNT pathway inhibitor, exhibit reduced numbers of crypts and villi, decreased villus size [40], as well as proliferation and differentiation defects—especially of secretory cell types including Paneth cells, goblet cells and enteroendocrine cells. This indicates that active WNT signaling controls intestinal epithelial cell proliferation, differentiation and renewal. In this regard, both LRP5 and LRP6 are expressed in epithelial cells at the bottom of adult intestinal crypts, where proliferation and differentiation are tightly regulated [41]. Notably, while conditional LRP5 or LRP6 deletion in intestinal epithelial cells does not alter intestinal architecture during late embryonic stages, conditional deletion of both LRP5 and LRP6 significantly perturbs the crypt–villus architecture, with decreased cell proliferation, reduced enteroendocrine cell numbers and altered enterocyte distribution along the crypt–villus axis [42]. These data suggest that LRP5 and LRP6 play compensatory roles during embryonic intestinal development. Interestingly, however, LRP5/6 and WNT/β-catenin are nonessential for the development of pseudostratified epithelium (E13.5, E14.5), the stage before the formation of columnar epithelium in typical villi [43].

## 3. LRP6 and Carcinoma Development

Given the importance of the WNT/β-catenin pathway in development, proliferation and differentiation, disruption of this pathway leads to the development of many diseases, including cancer. Deregulation of the pathway occurs at different levels, and mutations in genes encoding components of the destruction complex are central to tumor formation [44]. Additionally, some polymorphisms identified in the *LRP6* gene have been associated with different susceptibility to the development of cancers (Table 1).

The following sections provide an overview of current knowledge on the role and regulation of LRP6 in the development of epithelial cancers, including colorectal, liver, breast and pancreatic (as summarized in Figure 3).

### 3.1. Colorectal Cancer

#### 3.1.1. Molecular Biomarkers

With more than 1.8 million new cases and 881,000 deaths in 2018, colorectal cancers (CRCs) represent the third most diagnosed cancer worldwide and the second leading cause of death by cancer [55]. CRC incidence is higher in developed countries, and is increasing in transition countries. CRC is a heterogenous disease with distinct CRC subtypes identified and associated with unique clinical and molecular features [56]. Most sporadic CRCs arise in a multistep process, from mucosal hyperplasia to adenomas and carcinomas. Of these cancers, 70%–80% develop via conventional adenomatous polyps initiated by mutations activating the WNT/β-catenin pathway, commonly in the *APC* gene [57]. Mutually exclusive to APC, other common mutations of WNT pathway components include *CTNNB1*, *AXIN2*, *SOX9* and *RNF43* [58]. However, dysregulation of β-catenin signaling is not sufficient to generate the CRC phenotype. Chromosomal instability and additional somatic mutations in *KRAS*, *BRAF*, *PIK3CA* and/or inactivation of the *TP53*, *SMAD4* and *FBXW7* tumor suppressor genes collaborate to promote the progression of *APC*-mutant adenomatous lesions to CRC. Serrated adenocarcinomas, accounting for 20%–30% of CRCs, follow an alternative pathway independent of *APC* mutations, in which serrated polyps replace the traditional adenoma as the CRC precursor lesion [59]. This pathway involves early *BRAF* mutations, excess CpG island methylation and DNA microsatellite instability (MSI). Interestingly, increased WNT pathway activation is frequently observed in *BRAF*-mutated tumours due to loss-of-function mutations in *RNF43* [60]—an E3 ubiquitin ligase which mediates ubiquitination and degradation of the WNT receptor complex (Figure 1).

#### 3.1.2. Deregulation of LRP6 Expression and Function

Using whole-genome sequencing, three functional variants of LRP6 have been identified as novel candidate risk factors of early-onset CRC patients (Table 1) [47]. Besides that, in a systematic review of 11 publications, using sequence data from 863 familial CRC cases and 1604 individuals without CRC, Broderick et al. (2017) found no significant enrichment of LRP6 mutations in CRC [61,62]. Nevertheless, LRP6 is overexpressed in many CRC cell lines in comparison to normal epithelial cells, and 61% of malignant tissues isolated from sporadic CRC patients harbor higher LRP6 levels (Table 1) [53]. This increased LRP6 expression in CRC may result from hypermethylation of the Necdin promoter—a transcriptional repressor of the *LRP6* gene [63]. Aside from these observations, reduced expression of the LRP6 antagonist DKK1 has also been observed in CRC patients, and is inversely correlated with tumour grade and metastatic potential [64,65,66].

LRP6 phosphorylation is also increased in human colorectal tumours in comparison to healthy adjacent normal tissues (Table 1) [27,54]. This increased LRP6 phosphorylation correlates with tumour malignancy and staging as well as with poor prognosis of CRC, suggesting an important contribution of LRP6 activation in CRC progression and disease outcome [54]. In this regard, it has been shown that by inhibiting LRP6 expression, miR-195 down-regulates β-catenin signaling and transcription of target genes such as *RUNX2* and *VEGFa*, which in turn decrease CRC metastasis [67]. Additionally, the lncRNA-MALAT1, which is highly expressed in recurrent CRC, reduces the levels of miR-195, abrogating its inhibitory effect on LRP6 expression. Hence, MALAT1 was recently proposed to be a biomarker for predicting the recurrence and metastasis of CRC patients [67].

#### 3.1.3. Targeting LRP6 in CRC Models

Not surprisingly, LRP6 overexpression in CRC cells induces WNT/β-catenin activation, actin and microtubule remodeling and cell migration [54]. Conversely, LRP6 silencing—even in CRC cells with *APC* mutations—inhibits constitutive WNT pathway activation, indicating that LRP6 at least partially mediates the positive effect of *APC* inactivation on the WNT pathway [33]. These results were recently challenged by Chen and He (2019), who did not detect any modulation of WNT signaling activation and β-catenin levels in cell lines in which *APC* and *LRP6* and/or *LRP5* were knocked out by CRISPR/Cas9 genome editing [68]. However, single-cell analyses performed in isogenic human colonic epithelial cell lines confirmed that total and nuclear levels of β-catenin were increased upon LRP6 silencing, hence confirming an important role for LRP6 in WNT signaling of *APC*-mutated cells [69]. In this respect, LRP6 monoclonal antibodies targeting the WNT3 binding site significantly reduce proliferation and growth of tumorigenic *Apc^Min/+^* organoids (Table 2) [33]. Furthermore, a bispecific antibody (GSK3178022) to LRP6 that is capable of blocking stimulation by a range of WNT and R-spondin (RSPO) ligands in vitro potently delayed tumor growth in a patient-derived RSPO fusion model of CRC (Table 2) [70]. More recently, single-domain antibody fragments (VHH) against the WNT3-binding site have been shown to block *Rnf43* mutant intestinal organoid growth and survival, by depleting stem-like tumor cells (Table 2) [71]. Overall, these studies clearly indicate that LRP6 may represent a promising therapeutic target for CRC, in particular for WNT-hypersensitive tumours.

*KRAS* is the most commonly mutated oncogene in CRC. *KRAS* is mutated on codons 12, 13, 61, 117 or 146 in about 30–60% of patients with adenocarcinomas of the colon or rectum [72,73]. Our group has demonstrated in cellulo that the nuclear transcriptional activity of β-catenin is enhanced upon sustained oncogenic stimulation of normal intestinal epithelial cells (IECs) by KRAS (G12D) or its downstream effectors BRAF (V600E) and MEK1 (S218/222D) [27]. Importantly, the expression of a dominant-negative TCF4 mutant which inhibits β-catenin/TCF4 transactivation severely attenuated the IEC transformation induced by oncogenic MEK1 and reduced tumorigenic potential in mice [27]. These data suggest that KRAS/MAPK signaling uses the β-catenin pathway to induce IEC transformation. This is consistent with the recent observation that WNT signaling components were significantly enriched in KRAS-dependent CRC cells compared to KRAS-independent cells, even though comparable *APC* mutations occurred in both [74]. Notably, we recently observed that LRP6 was phosphorylated on serine-1490 and threonine-1572 in an MEK-dependent manner, in IEC transformed by oncogenic KRAS or BRAF, thus providing a mechanism integrating KRAS/MAPK and canonical WNT/β-catenin signaling during intestinal transformation (Figure 3A and Table 1) [27].

### 3.2. Liver Cancer

#### 3.2.1. Molecular Biomarkers

Liver cancer incidence is higher in the least-developed countries. Environmental risk factors for liver cancer include viral infections, alcohol intake, obesity and type 2 diabetes [55]. Hepatocellular carcinoma (HCC) represents 75–85% of liver cancers, with 50–83% of these cancers characterized by abnormal accumulation and distribution of β-catenin [75]. By contrast to CRC, HCC rarely exhibits *APC* mutations. However, *CTNNB1* or *AXIN1* mutations are frequently observed respectively in 20% and 55–100% of HCCs (100% of poorly differentiated HCCs) [75,76,77,78,79,80]. Thus, aberrant WNT/β-catenin pathway activation is likely involved in HCC tumorigenesis.

#### 3.2.2. Deregulation of LRP6 Expression and Function

Although no mutation has been reported in *LRP6*, 45–75% of HCC tumour samples harbor increased LRP6 expression in comparison to normal liver tissue. This increased LRP6 expression is associated with malignancy, poor prognosis and chemoresistance (Table 1) [48,49,50]. Accordingly, ectopic expression of a constitutively activated form of LRP6 in HCC cells promotes proliferation, invasion and migration, and increases tumour formation in immunodeficient mice [48]. Aside from these observations, it has recently been reported that high expression of LncRNA FOXD2-AS1, which targets the LRP6 inhibitor DKK1 mRNA, predicts poor prognosis in HCC patients. Mechanistically, by downregulating DKK1 expression, FOXD2-AS1 promotes the activation of WNT/β-catenin in HCC cell lines and enhances their tumorigenic properties (Figure 3B) [81]. Interestingly, LRP6 expression is also regulated by various miRNAs in HCC. MiR-202 and miR-126-3p expression is decreased in HCC tumour tissues, inversely correlating with LRP6 expression [82,83]. MiR-202 or miR-126-3p overexpression in HCC cells suppresses LRP6 expression, and dramatically inhibits HCC cell proliferative, angiogenic and invasive properties (Figure 2B) [82,83]. Moreover, by targeting LRP6, miR-1269a inhibits proliferation while inducing apoptosis of human HCC cells. Interestingly, a single nucleotide variant in miR-1269a, associated with increased susceptibility to HCC, was recently identified [84].

Hepatitis B (HB) viral infection is one of the major etiological factors for HCC, with chronic infection occurring in 50% of HCCs [85]. The HBV-expressed HBx protein is necessary not only for HBV infection but also for HCC development [86]. Intriguingly, a C-terminal truncated Hbx protein (HBx∆C) is found in 46–80% of HBV-infected HCCs [87,88]. A recent study reported that HBx∆C increased the expression of the lipid raft-associated protein caveolin-1, resulting in the activation of the WNT/β-catenin pathway [50]. More specifically, caveolin-1 acts by enhancing LRP6 protein stability, leading to the activation of the WNT pathway. Given the high incidence of HBV in HCC development, targeting LRP6 could be an efficient therapeutic strategy to limit tumour development in this context.

### 3.3. Breast Cancer

#### 3.3.1. Molecular Biomarkers

Breast cancers are a leading cause of cancer deaths, with a higher incidence in industrialized countries. Three main subtypes have been described, namely, luminal estrogen receptor (ER)-positive, HER2-enriched and triple negative breast carcinomas (TNBCs) that do not express ER, PR and HER2 [89]. Consequently, TNBC patients are unresponsive to targeted therapies, and treatment is limited to conventional chemotherapy. Therefore, there is an urgent need for new therapeutic targets specifically for these patients. Notably, nuclear β-catenin levels are increased in 40–60% of breast cancers, notably in TNBC subtypes associated with poor prognosis [90]. Interestingly, mice expressing an active form of β-catenin in the mammary gland develop basal-like tumours, suggesting a role for the WNT signaling pathway in breast tumour development. However, in contrast to what is observed in HCC or CRC, mutations of genes encoding components of the WNT/β-catenin pathway are rarely found in breast cancer [91,92]. Interestingly, expression of LRP6, but not LRP5, has been shown to define a new class of breast cancer subtypes, as LRP6 is overexpressed in 20–36% of breast carcinomas (Figure 3C and Table 1) [51].

#### 3.3.2. Targeting the Expression and Function of LRP6

LRP6 silencing in breast cancer cell lines impairs proliferation, anchorage-independent cell growth and tumorigenesis in xenograft assays. In addition, LRP6 is able to initiate mammary tumorigenesis, as seen with MMTV-LRP6 mice that develop mammary gland hyperplasia [93]. Notably, exogenous administration of a MESD peptide in a MMTV-WNT1 tumour mouse model suppressed tumor growth in vivo, likely through peptide binding to mature LRP6 at the cell surface and interference with ligand binding [51]. As discussed previously, LRP5/6-FZD receptor complexes are found in lipid rafts enriched with cholesterol and sphingolipids. Therefore, cholesterol depletion with methyl-β-cyclodextrins disrupts lipid rafts and decreases both LRP6 and β-catenin expression in TNBC-derived cells [94]. Finally, natural compounds isolated from plants (e.g., prodigiosin, silibinin, rottlerin, salinomycin and gigantol) all inhibit WNT/β-catenin activity and growth in TNBC cells by suppressing LRP6 expression and phosphorylation (Table 2) [95,96,97,98,99]. Moreover, niclosamide, a teniacide of the antihelminthic family, induces LRP6 degradation in breast cancer cell lines, resulting in increased cell apoptosis and decreased cell proliferation (Table 2) [100]. This emphasizes the possibility that LRP6 can represent a potential therapeutic target for breast cancers.

### 3.4. Pancreatic Ductal Adenocarcinoma

#### 3.4.1. Molecular Biomarkers

Pancreatic cancer is among the most devastating cancers, with a five-year survival rate less than 10% [55,101]. Pancreatic cancer incidence is explained by many risk factors, including smoking, alcohol intake, family history and genetic susceptibility with germline mutations in *BRCA1*, *BRCA2* and mismatch repair genes [101,102]. Pancreatic ductal adenocarcinoma (PDAC), the most common and aggressive form of pancreatic cancer, is mainly associated with the presence of oncogenic *KRASG12V* mutation [103,104]. Some studies highlight the promoting role of the WNT/β-catenin signaling pathway in KRAS-dependent PDAC initiation and progression [105,106]. Interestingly, 65% of PDAC tumor samples display increased β-catenin protein levels, which are associated with decreased serine 75/threonine 41 phosphorylation and β-catenin stabilization, but not with *CTNNB1* gene mutation [107]. Notably, most patient tumors exhibit enhanced β-catenin levels without modification of phosphorylation state, suggesting that other mechanisms are involved. That said, genomic analysis has revealed that 5–10% of PDAC patients harbor mutations in the *RNF43* gene [108,109,110,111]. Notably, these pancreatic tumours are highly dependent on WNT signaling (Figure 3D [112,113].

#### 3.4.2. Deregulation of LRP6 Expression and Function

LRP6 expression and activity are tightly regulated in PDAC. Increased LRP6 expression and phosphorylation associated with β-catenin nuclear accumulation have been observed in human-derived PDAC as well as murine KRAS-dependent pancreatic cancers (KPCs) [52]. More specifically, the microRNA miR-454, which targets LRP6 mRNA, is downregulated in PDAC (Figure 3D) and therefore its overexpression in pancreatic cancer cells impairs proliferation, angiogenic activity and metastatic potential [114,115]. Expression profiles of miRNA expressed in pancreatic cancers have correlated reduced miR-29c expression with poor prognosis [116]. Using TargetScan, LRP6, as well as FRAT2, FZD4 and FZD5, were identified as potential miR-29c targets. Not surprisingly, overexpression of miR-29c inhibits PDAC cell tumorigenicity and invasion, likely by direct inhibition of these WNT signaling proteins. On the other hand, LRP6 activity is also controlled by LINC01133, a long non-coding RNA upregulated in pancreatic cancers. Indeed, LINC01133 decreases DKK1 expression by inducing *DKK1* promoter methylation (Figure 3D) [117]. Therefore, LINCO1133 overexpression in pancreatic cancer cell lines increases β-catenin transcriptional activity and promotes proliferative, tumoral and invasive properties.

#### 3.4.3. Targeting the Expression and Function of LRP6

Interestingly, the *vitamin D receptor* gene has been recently identified as a determinant of survival in patients with pancreatic cancer [118]. In this regard, a negative correlation between patient’s vitamin D serum levels and pancreatic cancer has been observed, with vitamin D insufficiency associated with pancreatic cancer progression [118,119]. Notably, the vitamin D analogue calcipotriol, which displays anticancer properties in advanced pancreatic tumours, likely acts by reducing LRP6 protein levels (Table 2) [120,121]. In fact, treatment of pancreatic cancer cells with calcipotriol induces the expression of low-density lipoprotein receptor adaptor protein 1 (LDLRAP1), which interacts with tyrosine motifs of the LRP6 cytoplasmic tail, thereby inducing LRP6 clathrin-dependent endocytosis and subsequent lysosome-dependent degradation. Hence, by targeting LRP6 expression, vitamin D analogs act as therapeutic WNT inhibitors to inhibit PDAC tumor growth. In line with that, LRP6 silencing in human pancreatic cancer cell lines inhibits β-catenin/TCF transcriptional activity and cell proliferation [52].

### 3.5. Other Epithelial Cancers

Some evidence also suggests a role for LRP6 in the development of other epithelial cancers, including prostate, gastric, bladder, non-small-cell lung (NSCL) and papillary thyroid cancers. For instance, androgen-dependent LRP6 expression is necessary for prostate cancer cell growth [122]. As observed in breast cancer cells, silibinin, rottlerin, salinomycin and niclosamide were demonstrated to inhibit WNT/β-catenin activity and pancreatic cancer cell growth by suppressing LRP6 expression or phosphorylation (Table 2) [95,96,97,100]. Interestingly, treatment with recombinant MESD protein, a LRP5/6 chaperone protein able to bind to mature LRP5 and LRP6 on the cell surface, decreases LRP6 phosphorylation in prostate cancer cells, inhibiting their proliferation and tumorigenic potential (Table 2) [123,124]. On the other hand, infection of gastric epithelial cells with *Helicobacter pylori* or *Mycoplasma hyorhinis* induced rapid phosphorylation of LRP6 and downstream activation of Wnt/β -catenin signaling [125,126]. Additionally, curcumin and pantoprazole (a proton pump inhibitor) have demonstrated promising inhibitory effects on the growth of gastric adenocarcinoma (Table 2) [127,128]. In papillary thyroid carcinoma, miR126 and miR381-3p expression inhibits cell proliferation, migration/invasion and reduces tumour growth by decreasing LRP6 expression [129,130]. Regarding bladder cancer, ectopic expression of LRP6 in T24 cells promotes viability and invasion, while knockdown of LRP6 induces cell apoptosis [131]. Interestingly, the single-nucleotide polymorphism of LRP6 rs10743980 was recently associated with a decreased risk of developing bladder cancer [46]. Likewise, in a cohort of 500 non-small-cell lung cancer (NSCLC) patients, LRP6 rs10845498 was associated with a reduced risk of NSCLC and LRP6 rs6488507 with increased risk of NSCLC in tobacco smokers (Table 1) [45]. Further studies are however needed to elucidate the functional impact of LRP6 expression and activity in NSCLC.

## 4. LRP6: A Candidate for Targeted Therapy

Until now, the clinical trials that were conducted to inhibit the Wnt/β-catenin pathway mostly involved the use of porcupine inhibitors to block WNT secretion or the use of adenoviruses to overexpress DKK3, an antagonist of LRP5/6 (Table 3). Inhibitors of porcupine have demonstrated promising results in phase I clinical trials, inducing tumor regression in BRAF-mutant CRC, TNBC and pancreatic cancer patients (Table 3) [136,137,138]. However, given the crucial role of WNT signaling in the regulation of tissue homeostasis, global WNT inhibition may result in adverse or toxic effects. Interestingly, LRP6 alterations belong to inclusion criteria in clinical trials using inhibitors of porcupine in various cancers, highlighting the presumed critical role of LRP6 in cancer development [139]. Indeed, as mentioned in this review, upregulated LRP6 function in tumours is frequently associated with cell malignancy, poor prognosis and/or chemoresistance. Therefore, LRP6 protein clearly represents a pertinent actionable target for cancer therapy. In this regard, modified antibodies specifically targeting the extracellular domain of LRP6 were previously generated, and some of them seem very promising [9,33,70,71,132]. For instance, LRP6 monoclonal antibodies targeting the WNT3 binding site potently reduced the proliferation and growth of *Apc*-mutant intestinal tumoroids [33]. Nonetheless, the LPR6 extracellular domain is divided into two functional entities, P1E1P2E2 and P3E3P4E4, which respectively bind WNT1 and WNT3 glycoproteins. Therefore, the use of an antibody targeting the WNT3 binding site of LRP6 can sensitize cells to WNT1 ligands and vice versa, probably due to antibody-mediated LRP6 dimerization. Consequently, it becomes necessary to develop specific domain antibodies to selectively inhibit LRP6 activation by certain classes of WNTs while leaving the binding of other WNT ligands unchanged, limiting potential side effects. In this respect, Fenderico et al. (2019) recently used CIS display technology to identify single-domain antibody fragments (VHH) that bind the P3E3P4E4 region (WNT3-binding site). By inhibiting cellular responses to WNT3a but not those to WNT1, these anti-LRP5/6 VHHs efficiently block *Rnf43* mutant intestinal organoid growth and survival [71]. Thus, targeting specific regions of LRP6 ectodomain, especially the P3E3P4E4 region, may represent a promising strategy to reduce β-catenin-dependent signaling in tumours, without altering other WNT functions.

## 5. Conclusions

From these previous studies, it is clear that LRP6 is a key player in the activation of the WNT/β-catenin signaling pathway and, consequently, in the regulation of tissue homeostasis under physiological and pathological conditions. However, β-catenin-independent roles of WNT/LRP6 signaling have recently been reported [140]. How exactly these β-catenin-independent WNT/LRP6 signaling pathways contribute to tumour development remains to be determined. Furthermore, LRP6 is phosphorylated by mitogen-activated protein kinases such as ERK1/2, which transduce signals from the cell surface to the nucleus, thus affecting cell proliferation, differentiation and survival [25]. Notably, increased phosphorylation of LRP6 was observed in KRAS-transformed intestinal epithelial cells [27]. Hence, it will be relevant to determine whether inhibiting LRP6 function interferes with tumoral properties of human KRAS-mutated cancer cells. This would be a very important finding since KRAS is mutated in 80% of human PDAC, 40–50% of CRC and 30–50% of lung carcinomas and is very difficult to target. In the future, it will be necessary to fully characterize LRP6 activators and downstream effectors under specific cellular contexts. The elucidation of the LRP6 interactome may be one additional step to further understand LRP6 functions in cancer development.

## Figures and Tables

**Figure 1 cancers-11-01162-f001:**
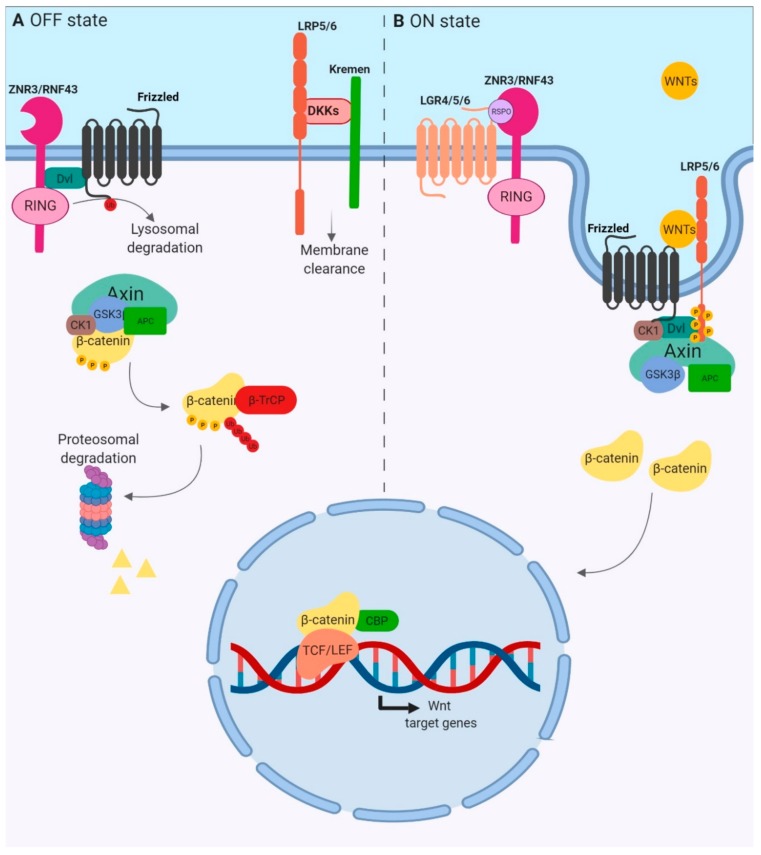
Overview of canonical WNT signaling. (**A**) OFF State. The absence of WNT ligands and the presence of antagonists such as DKK family members are associated with the inactive state of the pathway. The β-catenin destruction complex, formed by the scaffolding proteins AXIN and APC, and the kinases CK1α and GSK3, promotes the β-catenin phosphorylation, triggering its ubiquitination by the E3 ubiquitin ligase β-TrCP and its proteosomal degradation. (**B**) ON State. R-spondin (RSPO) and WNT ligands activate the pathway. RSPO binds both the ring finger proteins RNF43/ZNRF3 and the LGR receptor to form a ternary complex which is rapidly endocytosed. WNT glycoproteins are then able to bind Frizzled and LRP5/6 coreceptors, forming the WNT–FZD–LRP5/6 trimeric complex which recruits the scaffold protein Dishevelled (DVL) which itself polymerizes to bind AXIN. This promotes the formation of LRP6 signalosomes in which LRP6 is phosphorylated by CK1 and GSK3 kinases, releasing β-catenin, which accumulates and translocates into the nucleus to form an active transcription complex with TCF/LEF proteins.

**Figure 2 cancers-11-01162-f002:**
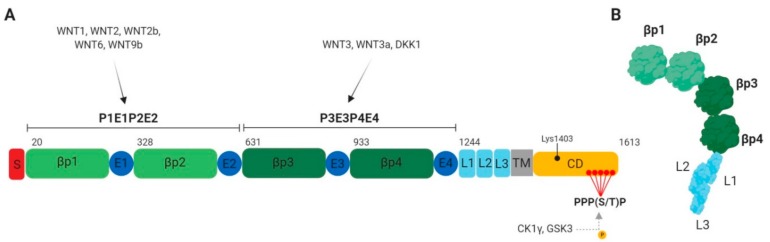
LRP6 structure. (**A**) Schematic representation of human LRP6 domains: βp, YWTD β-propeller domains; CD, cytoplasmic domain; E, EGF-like domains; L, LDLR type A repeats; S, signal peptide; TM, transmembrane domain. The five PPP(S/T)P phosphorylation motifs are shown in red. (**B**) LRP6 ectodomain in its resting state. In absence of ligands, LRP6 undergoes a large bending/unbending motion, preventing homodimer and signalosome formation [14].

**Figure 3 cancers-11-01162-f003:**
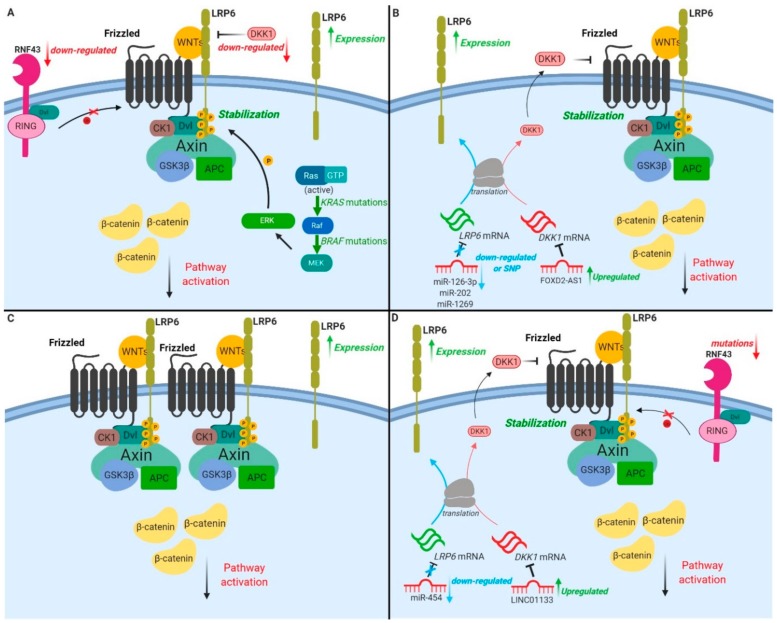
LRP6 and WNT/β-catenin pathway alterations in epithelial cancers. (**A**) In colorectal cancer (CRC), LRP6 overexpression is observed in 61% of malignant tissues isolated from patients. (i) Mutations in the E3 ubiquitin-protein ligase RNF43, (ii) downregulation of the antagonist DKK1 and (iii) overactivation of the RAS/MAPK pathway result in the recruitment of degradation complex components at the membrane and WNT/β-catenin pathway overactivation. (**B**) In hepatocellular carcinomas (HCCs), LRP6 is overexpressed in 45%–75% of HCC tumours. Downregulation of miR-126-3p, miR-202 and single-nucleotide polymorphism of miR-1269 result in increased LRP6 mRNA stability and protein levels. The long non-coding RNA FOX2-ASI, upregulated in HCC, targets and downregulates *DKK1* mRNA (LRP6 inhibitor), thus increasing LRP6 membrane levels. (**C**) In breast cancers (BCs), LRP6 overexpression is observed in 20%–36% of carcinomas, and β-catenin nuclear accumulation is reported in 40%–60% of cases. (**D**) In pancreatic ductal adenocarcinoma (PDAC), (i) mutations of *RNF43* are observed in 5%–10% of PDAC patients, (ii) down-regulation of miR-454 and (iii) upregulation of LINC01133—a long non-coding RNA which induces *DKK1* promoter methylation (decreasing DKK1 expression)—all explain increased LRP6 expression and subsequent increased β-catenin-dependent signaling.

**Table 1 cancers-11-01162-t001:** LRP6 functional variants and expression status in distinct cancers. Some single-nucleotide polymorphisms (SNPs) and mutations in the *LRP6* gene have been associated with increased or decreased risk of cancer development.

Variants and Expression Status	Cancer	Risk and Prognosis	References
LRP6 rs6488507	Non-small-cell lung cancer (NSCLC)	Increases the risk of NSCLC in tobacco smokers	[45]
LRP6 rs10845498	Lung squamous cell carcinoma (SCC)	Associated with a reduced risk of SCC	[45]
LRP6 rs10743980	Bladder cancer	Associated with a decreased risk of bladder cancer	[46]
LRP6 rs141458215 (p.T867A) p.N789S p.W239L	Colorectal cancer (CRC)	Novel candidate risk factor for early onset of CRC	[47]
Overexpression	Hepatocellular carcinoma (HCC)	Associated with malignancy, poor prognosis and chemoresistance	[48,49,50]
Overexpression	Breast cancer	Defines a new class of breast cancer subtype	[51]
Overexpression Increased activity	Pancreatic ductal adenocarcinoma (PDAC) KRAS-dependant pancreatic cancer	Associated with tumour progression	[52]
Overexpression Increased activity	Colorectal cancer (CRC)	Associated with malignancy and poor prognosis	[27,53,54]

**Table 2 cancers-11-01162-t002:** List of therapeutic reagents targeting LRP6 in various cancers. Different antibodies, small molecules and peptides have been shown to exert anticancer properties by directly or indirectly inhibiting LRP6 function.

Therapeutic Reagents	Categories	Targeting Mechanism	Cellular Effects	Cancer Type	References
BpAb A7/B2	Bispecific Antibody	Competition for WNT binding on LRP6	Blocks both Wnt1- and Wnt3a-mediated β-catenin signaling and xenograft tumour growth	Breast cancer	[9]
mAb7E5	Antibody	Competition for WNT binding on LRP6	Decreases nuclear β-catenin localisation and activity; Reduces proliferation and growth of tumorigenic Apc-mutated organoids	Colorectal cancer	[33]
GSK3178022	Bispecific Antibody	Competition for WNTs binding on LRP6	Decreases TCF/LEF transcriptional activity; Reduces tumor growth of patient-derived colorectal xenografts (PDX)	Colorectal cancer	[70]
YW211.31.57	Antibody	Competition for WNT3 binding	Decreases TCF/LEF transcriptional activity	Breast cancer	[132]
YW210.09	Antibody	Competition for WNT1 binding	Decreases TCF/LEF transcriptional activity; Inhibits MMTV-Wnt1 xenograft tumour growth	Breast cancer	[132]
VHH	Single antibody fragment	Competition for WNT3-binding on LRP6	Abrogates cellular response to WNT3a, but not to WNT1 Blocks Rnf43/Znrf3 mutant intestinal organoid growth	Colorectal cancer	[71]
Calcipotriol	Small molecule Vitamin D analog	Induces LDLRAP1 expression which interacts with LRP6	Induces clathrin-dependent endocytosis and lysosome-dependent LRP6 degradation	Pancreatic cancer	[121]
Curcumin	Small molecule	Decreases LRP6 expression and phosphorylation	Increases cell apoptosis; Reduces cell proliferation, colony formation and invasion; Decreases tumour growth	Gastric adenocarcinoma	[127]
Pantoprazole	Small molecule (proton pump inhibitor)	Decreases LRP6 phosphorylation	Decreases cancer cell growth and invasion properties; Increases cancer cell apoptosis	Gastric adenocarcinoma	[128]
Prodigiosin	Small molecule	Decreases LRP6 phosphorylation	Decreases cancer cell viability, proliferation, migration and invasion properties; Increases cell apoptosis; Reduces breast xenograft tumour growth	Breast cancer	[99]
MESD	Peptide	LRP6 binding	Tumour growth reduction in MMTV-WNT1 model	Breast cancer	[51]
MESD	Peptide	LRP6 binding	Decreases LRP6 phosphorylation and reduces cancer cell proliferation; Reduces prostate xenograft tumour growth	Prostate cancer	[123,124]
Silibinin, Rottlerin, Gigantol	Small molecules	Reduce LRP6 expression and phosphorylation	Growth inhibition of cancer cell lines	Prostate and Breast cancer	[95,97,98]
Salinomycin	Small molecule	Reduces LRP6 expression	Inhibition of cancer cell proliferation	Breast cancer, Chronic lymphocytic leukemia	[96,133]
Niclosamide	Small molecule (Antihelminthic)	Reduces LRP6 expression and phosphorylation Induces LRP6 degradation	Induction of cancer cell apoptosis Inhibition of cancer cell proliferation	Prostate cancer, Breast cancer, Ovarian cancer	[100,134]
		Reduces LRP6 expression and phosphorylation	Induction of cancer cell apoptosis Inhibition of cell proliferation, migration and angiogenesis	Retinoblastoma	[135]

**Table 3 cancers-11-01162-t003:** Clinical trials targeting effectors of WNT/β-catenin pathway. Given the importance of the WNT/β-catenin pathway in cancer development, effectors of this pathway such as porcupine and LRP6 are specifically targeted in Phase 1 and 2 clinical trials in various cancers. Ad: adenovirus, DKK3: Dickkopf-related protein 3, RAF: rapidly accelerated fibrosarcoma, PD-1: Programmed cell death 1, EGFR: epidermal growth factor receptor.

Intervention	Target	Condition or Disease	Sponsor	Phase	Submitted Date	Status	Identifier
Drug: LGK974 (Other names: WNT974, porcupine inhibitor) Biological: PDR001 (PD-1 monoclonal antibody)	Porcupine PD-1	Pancreatic Cancer, BRAF Mutant CRC, Melanoma, Breast cancer (TNBC), Squamous Cell Cancers: head, neck, cervical, esophageal, lung	Novartis Pharmaceuticals	Phase 1	4 May, 2011	Recruiting	NCT01351103
Drug: WNT974 (porcupine inhibitor) Drug: LGX818 (RAF inhibitor) Biological: cetuximab (EGFR antibody)	Porcupine RAF EGFR	Metastatic BRAF Mutant CRC	Array BioPharma	Phase 1 Phase 2	6 October, 2014	Completed	NCT02278133
Drug: WNT974 (porcupine inhibitor)	Porcupine	Head and Neck Squamous Cell Cancer	University of Michigan Rogel Cancer Center	Phase 2	5 January, 2016	Withdrawn	NCT02649530
Biological: Ad-REIC/Dkk-3 (↑DKK3 expression)	Complex receptor (LRP6)	Prostate cancer	Momotaro-Gene	Phase 1	3 September, 2010	Withdrawn (Suspended)	NCT01197209
Drug: Ad5-SGE-REIC/Dkk3 (↑DKK3 expression)	Complex receptor (LRP6)	Prostate cancer	Momotaro-Gene	Phase 1 Phase 2	21 August, 2013	Active, not recruiting	NCT01931046
Drug: niclosamide (Antihelminthic)	Complex receptor (LRP6)	Colon cancer	M. Morse MD	Phase 1	10 February, 2016	Recruiting	NCT02687009

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
