# Peer review of "A Role for the WNT Co-Receptor LRP6 in Pathogenesis and Therapy of Epithelial Cancers"

_cancers, 2019, doi:10.3390/cancers11081162_

Round 1

Reviewer 1 Report

Jennifer Raisch et al., summarizes current knowledge regarding the biological function and regulation of LRP6 in the development of various cancer types. In my opinion, its publication may be appropriate after addressing several points. If authors satisfactorily address this concern, this work would greatly contribute to cancer research.

They need to add more in detail current information for the functional roles of receptor-related protein 6 (LRP6) in tumor development of various cancer types regarding its biological functions (growth, metastasis, recurrence, prognosis, drug resistance..etc) and its molecular mechanism (regulatory pathway, downstream signaling pathways, and target gene)

They also need to add some information about function and regulation of LRP6 in the development of non-solid tumor, including hematological tumor.

They also need to add some information about the roles of LRP6 in the drug resistance and recurrence in multiple cancer types.

Please add more in detail information about the current therapeutic approaches for targeting LRP6 in multiple cancer types and further information for their therapeutic effects, limitation, alternative.

Please add future prospective for LRP6 targeting therapeutic approaches in various cancer types.

Author Response

We sincerely thank the referee for the constructive comments about our work. Here is a detailed point-by-point response to the comments. 

Response to reviewer #1

“The authors need to add more in detail current information for the functional roles of receptor-related protein 6 (LRP6) in tumor development of various cancer types regarding its biological functions and its molecular mechanism.” First, we have to mention to the reviewer that we had decided to focus the review on cancers from epithelial origin and for which, sufficient information about LRP6 expression/function is described in the literature. Thus, to avoid any confusion, we have changed the title of the review which is now “A role for the WNT co-receptor LRP6 in pathogenesis and therapy of epithelial cancers”. We also modified the abstract accordingly and mentioned that the review “summarizes current knowledge regarding the biological function and regulation of LRP6 in development of epithelial cancers, especially colorectal, liver, breast and pancreatic cancers”. They also need to add some information about function and regulation of LRP6 in the development of non-solid tumor, including hematological tumor.” We understand the reviewer concern but as mentioned before, the review is focused on the role and regulation of LRP6 in colorectal, liver, breast and pancreatic cancers. Moreover, in the revised manuscript, we added a new section describing the little known about LRP6 expression/function in other epithelial cancers such as prostate, gastric, bladder, lung and papillary thyroid cancers (see new section 3.5). “They also need to add some information about the roles of LRP6 in the drug resistance and recurrence in multiple cancer types”. To our knowledge, very few information is known about the specific role of LRP6 in drug resistance and cancer recurrence. Nonetheless, we have added few informations that we have found in section 3.1.2 and 3.2.2 (see in red). Please add more in detail information about the current therapeutic approaches for targeting LRP6 in multiple cancer types and further information for their therapeutic effects, limitation, alternative”. The current status of targeting LRP6 is further clarified in the revised manuscript. Also, we have now included a table (new Table 2) summarizing the therapeutic reagents targeting LRP6, the cancer type, as well as the cellular and/or clinical outcome if applicable.Besides that, we have included an additional table (new Table 3) summarizing the clinical trials with inhibitors/drugs targeting the Wnt/b-catenin pathway alone or in combination therapy. Please add future prospective for LRP6 targeting therapeutic approaches in various cancer types”.As suggested by the reviewer, we have added some future prospective for LRP6 targeting in section 4 (see in red).

Reviewer 2 Report

!. The authors provide very detail information regarding LRP6 function in WNT pathways, the mechanism of LRP6 in tissue homeostasis, and its role and the possible mechanisms for the carcinogenesis in breast, liver, coloreactal , and pancreatic ductal adenocarcinoma.. 

2. They give a reasonable evaluation whether LRP6 can be serve as a drug target.  

3. This manuscript provides enough information regarding LRP6 function from scientific point of view. The detail figure to illustrate the function domains and post-modification sites of LRP6 might make this manuscript more informative.    

Author Response

We sincerely thank the referee for his/her very positive comments about our work.

Reviewer 3 Report

This paper discussed the LRP6 function and regulation in cancer development with sufficient depth and width. The writing is cohesively reasoned but require better structures. The clinical status of targeting LRP6 need to be discussed in more detail. In addition, it would be helpful if the author would provide more figures and tables that navigate the readers to appreciate the content of the paper.

Major:

1.    Section 2.1 Structure of LRP6: It would be great if the author could provide a figure of the structure of LRP6, and a table outlining the position of common LRP6 mutations with the results of these mutations discussed in the main text.

2.    Section 2.2 Regulation: The author describes the post-translational regulation of LRP6 folding, surface aggregation, and recycling extensively. Would it be possible to discuss the translational aspects of LRP6 regulation? Is there evidences that the functional alteration of LRP6 is mainly driven by post-translational regulation? Otherwise, it would be more precise if the author would narrow the title of this section to “post-translational regulation”.

3.    Section 2.3. The structure of the section is slightly confusing. The author could separate this part into a) the embryonic development and b) the intestinal epithelial homeostasis in adults with the evidence of LRP5/6 expression in adults (Ref. 39 and 40).

4.    Figure2: It would be better if the author could incorporate the potential targeting strategies of WNT signaling in corresponding cancers.

5.    Section 4: The current status of targeting LRP6 could be further clarified. It would be great if the author could provide a table listing the current therapeutic reagents targeting LRP6, including the reagent name, categories (antibody/ small molecules, etc.), targeting mechanism, cancer type, as well as the clinical outcome if applicable.

6.    The abstract of the paper could match with the flow of main-content more cohesively.

Minor:

The author uses CRC to compare with HCC on the current pg.6. Also, to connect with the discussion of the intestinal epithelial homeostasis, it would be more logical to move the discussion of CRC to first cancer before the breast cancer and match the sequences of cancer stated in the abstract.

An incomplete sentence on page 10: “Notably, increased phosphorylation of LRP6 in KRAS-transformed intestinal epithelial cells.”

Author Response

We sincerely thank the referee for the constructive comments about our work. Here is a detailed point-by-point response to the comments.

Major 

Section 2.1 Structure of LRP6: It would be great if the author could provide a figure of the structure of LRP6, and a table outlining the position of common LRP6 mutations with the results of these mutations discussed in the main text.  Following the reviewer’s suggestion and in an effort to be more informative in the review, we have added a new figure (new Figure 2) in the revised manuscript illustrating the structure of LRP6 with the extracellular, transmembrane and intracellular domains. The known phosphorylation sites are also indicated. Section 2.2 Regulation: The author describes the post-translational regulation of LRP6 folding, surface aggregation, and recycling extensively. Would it be possible to discuss the translational aspects of LRP6 regulation? Is there evidences that the functional alteration of LRP6 is mainly driven by post-translational regulation? Otherwise, it would be more precise if the author would narrow the title of this section to “post-translational regulation”. As suggested by the reviewer, we changed the title of this section 2.2 for “Post-translational maturation and regulation”. Consequently, the paragraph describing the role of phosphorylation in the regulation of LRP6 function was also included in this section and a new section entitled “2.3 Endocytosis” was created. Section 2.3. The structure of the section is slightly confusing. The author could separate this part into a) the embryonic development and b) the intestinal epithelial homeostasis in adults with the evidence of LRP5/6 expression in adults (Ref. 39 and 40). As suggested by the reviewer, we separated this section in two parts in the revised manuscript. Figure2: It would be better if the author could incorporate the potential targeting strategies of WNT signaling in corresponding cancers. As suggested by the reviewer, we have now incorporated the targeting strategies with each corresponding cancer (see in red). Section 4: The current status of targeting LRP6 could be further clarified. It would be great if the author could provide a table listing the current therapeutic reagents targeting LRP6, including the reagent name, categories (antibody/ small molecules, etc.), targeting mechanism, cancer type, as well as the clinical outcome if applicable.  This suggestion of the reviewer is really pertinent. We have now included such table in the revised manuscript (see new Table 2). This new table summarizes the therapeutic approaches targeting LRP6 in cancers, mostly epithelial cancers. The abstract of the paper could match with the flow of main-content more cohesively. The abstract was revised to be more in line with the flow of content of the manuscript.

Minor:

The author uses CRC to compare with HCC on the current pg.6. Also, to connect with the discussion of the intestinal epithelial homeostasis, it would be more logical to move the discussion of CRC to first cancer before the breast cancer and match the sequences of cancer stated in the abstract. We have followed the reviewer's suggestion and revised accordingly the abstract.

An incomplete sentence on page 10: “Notably, increased phosphorylation of LRP6 in KRAS-transformed intestinal epithelial cells.” This sentence was completed in the revised manuscript.

Round 2

Reviewer 1 Report

All raised issues are properly addressed

Author Response

Thank you very much again for your constructive comments.

Reviewer 3 Report

The authors successfully addressed all of the reviewers' comments. Overall, the manuscript meets the publication requirement in the present form.

Author Response

(The authors gave the same response as above.)
